# Deep convolutional and conditional neural networks for large-scale genomic data generation

**Burak Yelmen** [1,2]*, **Aurélien Decelle** [1,3], **Leila Lea Boulos** [1,4], **Antoine Szatkownik** [1], **Cyril Furtlehner** [1], **Guillaume Charpiat** [1], **Flora Jay** [1]

**1** Université Paris-Saclay, CNRS, INRIA, LISN, Paris, France, **2** University of Tartu, Institute of Genomics, Tartu, Estonia, **3** Universidad Complutense de Madrid, Departamento de Física Teórica, Madrid, Spain, **4** Université d'Évry Val-d'Essonne, Évry-Courcouronnes, France

\* burakyelmen@gmail.com

**Data Availability Statement:** All the relevant data and code is publicly available at https://gitlab.inria.fr/ml_genetics/public/artificial_genomes. Another example of RBM code applied to images can be

## Abstract

Applications of generative models for genomic data have gained significant momentum in the past few years, with scopes ranging from data characterization to generation of genomic segments and functional sequences. In our previous study, we demonstrated that generative adversarial networks (GANs) and restricted Boltzmann machines (RBMs) can be used to create novel high-quality artificial genomes (AGs) which can preserve the complex characteristics of real genomes such as population structure, linkage disequilibrium and selection signals. However, a major drawback of these models is scalability, since the large feature space of genome-wide data increases computational complexity vastly. To address this issue, we implemented a novel convolutional Wasserstein GAN (WGAN) model along with a novel conditional RBM (CRBM) framework for generating AGs with high SNP number. These networks implicitly learn the varying landscape of haplotypic structure in order to capture complex correlation patterns along the genome and generate a wide diversity of plausible haplotypes. We performed comparative analyses to assess both the quality of these generated haplotypes and the amount of possible privacy leakage from the training data. As the importance of genetic privacy becomes more prevalent, the need for effective privacy protection measures for genomic data increases. We used generative neural networks to create large artificial genome segments which possess many characteristics of real genomes without substantial privacy leakage from the training dataset. In the near future, with further improvements in haplotype quality and privacy preservation, large-scale artificial genome databases can be assembled to provide easily accessible surrogates of real databases, allowing researchers to conduct studies with diverse genomic data within a safe ethical framework in terms of donor privacy.

## Author summary

Generative modelling has recently become a prominent research field in genomics, with applications ranging from functional sequence design to characterization of population

found at https://github.com/AurelienDecelle/TorchRBM.

**Funding:** This work was funded by the Agence Nationale de la Recherche through grant ANR-20-CE45-0010-01 RoDAPoG (B.Y., L.B., A.S., C.F., G. C., F.J.); the Comunidad de Madrid and the Complutense University of Madrid (Spain) through the Atracción de Talento programs (Refs. 2019-T1/TIC-13298), the Banco Santander and the UCM (grant PR44/21-29937), and the Ministerio de Economía y Competitividad, Agencia Estatal de Investigación and Fondo Europeo de Desarrollo Regional (FEDER) (Spain and European Union) through grant PID2021-125506NA-I00 (A.D.); Labex DigiCosme (project ANR-11-LABEX-0045-DIGICOSME) operated by ANR as part of the program "Investissement d'Avenir" Idex Paris-Saclay (ANR-11-IDEX-0003-02) (L.B.). The funders had no role in study design, data collection and analysis, decision to publish, or preparation of the manuscript.

**Competing interests:** The authors have declared that no competing interests exist.

structure. We previously used generative neural networks to create artificial genome segments which possess many characteristics of real genomes but these segments were short in size due to computational requirements. In this work, we present novel generative models for generating artificial genomes with larger sequence size. We test the generated artificial genomes with multiple summary statistics to assess the haplotype quality, overfitting and privacy leakage from the training dataset. Our findings suggest that although there is still room for improvement both in terms of genome quality and privacy preservation, convolutional architectures and conditional generation can be utilised for generating good quality, large-scale genomic data. In the near future with additional improvements, large-scale artificial genomes can be used for assembling surrogate biobanks as alternatives to real biobanks with access restrictions, increasing data accessibility to researchers around the globe.

## Introduction

Machine learning is an important staple in modern genomic studies. There have been numerous applications in demographic inference [1], detecting natural selection [2], genome-wide association studies [3] and functional genomics [4], many of which became state of the art [5, 6]. In the recent few years, generative machine learning approaches for the genomics field have also begun to gain research interest thanks to algorithmic advances and widespread availability of computational resources [7–11]. Broadly speaking, generative machine learning involves the utilisation of generative models which are trained to model the distribution of a given dataset so that new data instances with similar characteristics to the real data can be sampled from this learned distribution. Especially since the introduction of generative adversarial networks (GANs) in the preceding decade [12], generative modelling has become a widely-researched subject with a diverse scope of applications such as image and text generation [13, 14], dimensionality reduction [15] and imputation [16].

The amount of genetic data, both in sequence and SNP array formats, is now increasing at an unmatched rate, yet its accessibility remains relatively low. The main reason for this is the crucial protocols prepared to protect the privacy of donors. Genetic data belongs to a special category similar to private medical records in the General Data Protection Regulation (GDPR) [17] and is defined as protected health information in the Health Insurance Portability and Accountability Act (HIPAA) [18]. Although these protective measures are vital, they create an accessibility issue for researchers who must go through these protocols and, in many circumstances, might have to commit to collaborations to conduct research or simply test ideas. In our previous study, we introduced the concept of high quality artificial genomes (AGs) created by generative models as a possible future solution for this problem and as an alternative to other methods such as differential privacy [19] and federated learning [20]. We demonstrated that AGs can mimic the characteristics of real data such as allele frequency distribution, haplotypic integrity and population structure, and can be used in applications such as genomic imputation and natural selection scans [21]. However, our previous models lacked the capacity to generate large-scale genomic regions due to the computational requirements caused by the high number of parameters defining the fully connected architectures. In this study, we present two novel implementations better adapted to large sequential genomic data: (i) generative adversarial networks with convolutional architecture and Wasserstein loss (WGAN) [22], and (ii) restricted Boltzmann machines with conditional training (CRBM) [23] used together with an out-of-equilibrium procedure [24]. In more detail, we implemented a WGAN with gradient

penalty (WGAN-GP) [25] which involved a deep generator and a deep critic architecture, multiple noise inputs at different resolutions for the generator [26], trainable location-specific vectors for preserving the positional information [27], residual blocks to prevent vanishing gradients [28] and packing for the critic to eliminate mode collapse [29]. For the CRBM, we used the more efficient out-of-equilibrium training scheme differently from our previous study and developed a novel procedure for conditionally training multiple RBMs based on shared genomic regions [30]. We assessed the AGs generated by these models with multiple statistics measuring data quality and possible privacy leakage. First, we compared the new models with the ones from [21] using the same 10,000 SNPs extracted from the 1000 Genomes human dataset [31] and then trained these models with a larger 65,535-SNP dataset to generate long sequence AGs. We performed multiple tests to evaluate (i) the quality of the generated haplotypes via allele frequency spectrum, linkage disequilibrium, population structure and complex haplotypic integrity analyses such as 3-point correlations and distribution of k-mer motifs; and (ii) privacy preservation via distance-based metrics, membership inference attacks and overfitting/underfitting scores.

## Materials and methods

### Data

The two sets of 1000 Genomes [31] data we used include: (i) 10,000 SNPs from chromosome 15 (between 27,379,578—29,625,035 base pairs, $\sim$ 2 megabase pairs) identical to the ones picked by [21], and (ii) 65,535 SNPs from chromosome 1 between 534,247—81,813,279 base pairs, $\sim$ 80 megabase pairs) within the Omni 2.5 genotyping array framework. Further downsampling of the array framework was performed to create a dataset with a reasonable SNP number for faster training trials and the specific number of 65,535 SNPs was decided to be in the form of (2n—1) for easy implementation of convolutional scaling. For the 10,000-SNP dataset, we used padding with zeros to match the (2n—1) form. Both datasets have the same 2504 individuals and 5008 phased haplotypes used for training the models. The data format is the same as [21] where the rows are phased haplotypes and columns are positions which hold alleles represented by 0 (reference) and 1 (alternative).

### WGAN implementation

We implemented a Wasserstein GAN with gradient penalty (WGAN-GP) consisting of a critic which estimates the earth mover's distance between real and generated data distributions, and a generator which generates new genomic data from Gaussian noise (Fig 1). Unlike the discriminator in naive GAN which performs a classification task, the critic provides a "realness" score (an approximation of Earth-mover's distance) for generated and real samples. WGAN objective function to be minimized by the generator and maximized by the critic is as follows:

$$E_x[C(x)] - E_z[C(G(z))]$$

where $C$ is the critic, $G$ is the generator, $x$ is real data point and $z$ is Gaussian noise. The critic function $C$ must be Lipschitz continuous; thus, the original study relied on weight clipping to enforce an upper bound for the gradient [22]. However, we designed a WGAN with gradient penalty (WGAN-GP) rather than weight clipping, as GP was shown to be a better alternative [25]. In our implementation, the critic uses convolution layers whereas the generator uses convolution and transposed convolution layers (Fig 1A). Both the generator and the critic have trainable location-specific vectors as additional channels at every block except for the residual blocks. These vectors, similar to the ones integrated by [27] in their autoencoder, consist of

a)

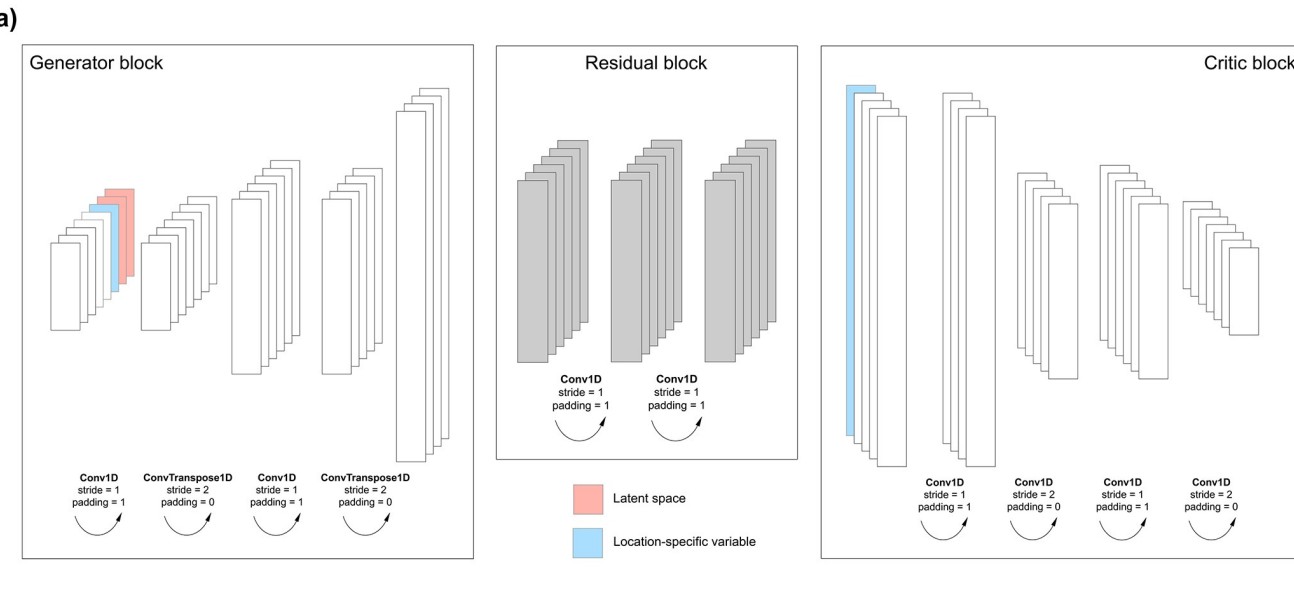

b)

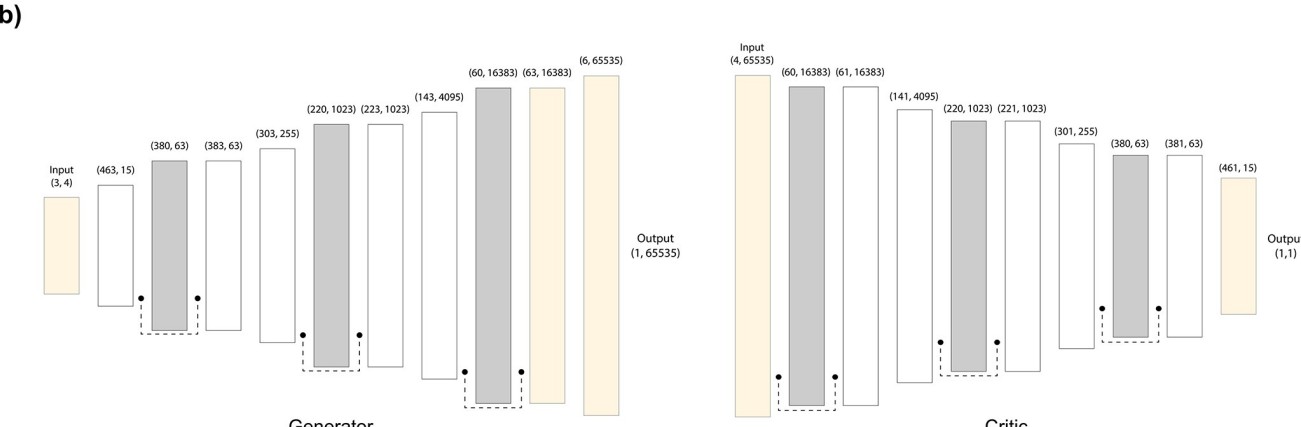

**Fig 1. Wasserstein GAN (WGAN) model for the 65,535-SNP dataset. a)** Representation of the generator, critic and residual blocks. Channel dimensions are not proportional and do not reflect the real implementation. The generator block has one trainable location-specific variable (blue) and two latent space vectors (red) as additional channels concatenated to the input. The critic block has one trainable location-specific variable (blue) as an additional channel concatenated to the input. **b)** Architecture of the WGAN model. White rectangles correspond to generic generator and critic blocks whereas grey rectangles correspond to generic residual blocks. Numbers in parentheses above blocks show channels and length, respectively (C, L). Dotted connections are residual connections where the input value is added to the output value of the block before passing to the next block. Yellow input and output blocks differ from the generic ones for proper dimension adjustments.

trainable variables that allow the models to preserve positional information which would otherwise diminish due to the invariance of convolution operations. In addition to this, the generator has two noise channels at every block. This gives the generator flexibility to decide at which depth the mapping to latent space can occur. We also implemented "packing" to overcome mode collapse by adjusting the discriminator to take multiple samples as input [29]. In our implementation, the input sample number for the discriminator was set to 3 intuitively, to match the 3 main population modes (Africa, West Eurasia, Asia) observable in the 1000 Genomes data. The initial length of the input for the generator is 4, which is gradually transformed into features of larger sizes until it reaches the sequence size of 65,535 or 16,383 in the output (Fig 1B).

The generator layers are followed by batch normalisation and leaky ReLU activation (alpha = 0.01) except for the final layer which has a sigmoid activation. The critic layers are followed by instance normalisation and leaky ReLU activation (alpha = 0.01) except for the final layer which is a fully connected layer with no activation. We used Adam optimizer to train both the generator and the critic with a learning rate of 0.0005 and $\beta_1, \beta_2 = (0.5, 0.9)$. $\beta_1$ was set to 0.5 as suggested by [13]. For each batch training of the generator, the critic was trained 10 times as suggested by the authors of the original WGAN [22]. We assessed the outputs of the generator at each epoch during training via PCA. We stopped training when generated and real genome clusters visually overlapped in PC space (components 1 to 4). In the case of 65,535-SNP data, this initial training was not sufficient to reach a good overlap of higher degree principal components, thus, we performed a second brief training (up to 200 epochs) with 10-fold lower generator learning rate (0.00005). Our WGAN architecture for the 10,000-SNP data was conceptually the same as for the 65,535-SNP data, but shallower with fewer blocks. All WGAN models were coded with python-3.9 and pythorch-1.11 [32]. The detailed python scripts can be accessed at https://gitlab.inria.fr/ml_genetics/public/artificial_genomes.

## RBM implementation

An RBM is a generative stochastic neural network [33] defined as a probability distribution over a set of visible units, $\underline{s}$, representing SNPs in our case, and hidden units, $\underline{\tau}$, where both type of variables interact via a pairwise weight matrix $\underline{w}$ and the local biases $\underline{\theta}$ and $\underline{\eta}$ help to adjust the mean value of each unit:

$$p(\underline{s}, \underline{\tau}) = \frac{1}{Z} \exp\left(\sum_{ia} w_{ia} s_i \tau_a + \sum_i \theta_i s_i + \sum_a \eta_a \tau_a\right)$$

In this study, we used binary units {0, 1} for both the visible and hidden nodes, and the number of hidden units was chosen to be about the same order as the number of visible ones. The likelihood of such model is given by:

$$L = \sum_{m=1}^{M} \left[\sum_i \theta_i s_i^{(m)} + \sum_a \log(1 + \exp(\sum_i w_{ia} s_i^{(m)} + \eta_a))\right] - M \log(Z)$$

where the index $m$ is indexing the samples of the dataset ($M$ being the total number of samples) and $Z$ is the normalization constant, or partition function, of the probability distribution.

Learning an RBM consists in maximising this likelihood using gradient descent in order to optimize the weights and biases $w$, $\theta$ and $\eta$. In our implementation, the training was based on the out-of-equilibrium method [24, 34]. The main difference with the more conventional learning is that the sampling which is done to compute the correlation of the model is performed in a very precise way: a random initial condition is chosen amongst a certain probability distribution $p_0(x)$ (kept fixed during the learning), and a fix (all along learning) number of Monte Carlo steps is done during the training. When using this particular training procedure, in order to sample new data, it is enough to generate the Monte Carlo chains following the same dynamical process: same initial conditions $p_0(x)$ and same number of MC steps. In the provided experiments, the initial conditions were chosen uniformly at random (each unit having equal probability to be 0 or 1) and the number of MC steps was either 100 (for the 65,535-SNP dataset) or 200 (for the 10,000-SNP dataset) but the qualitative results were mostly not affected. The learning rate of the model was chosen such that the eigenvalues of the learning weight matrix are smoothly increasing during the first epochs from almost zero to values of $\sim O(1)$.

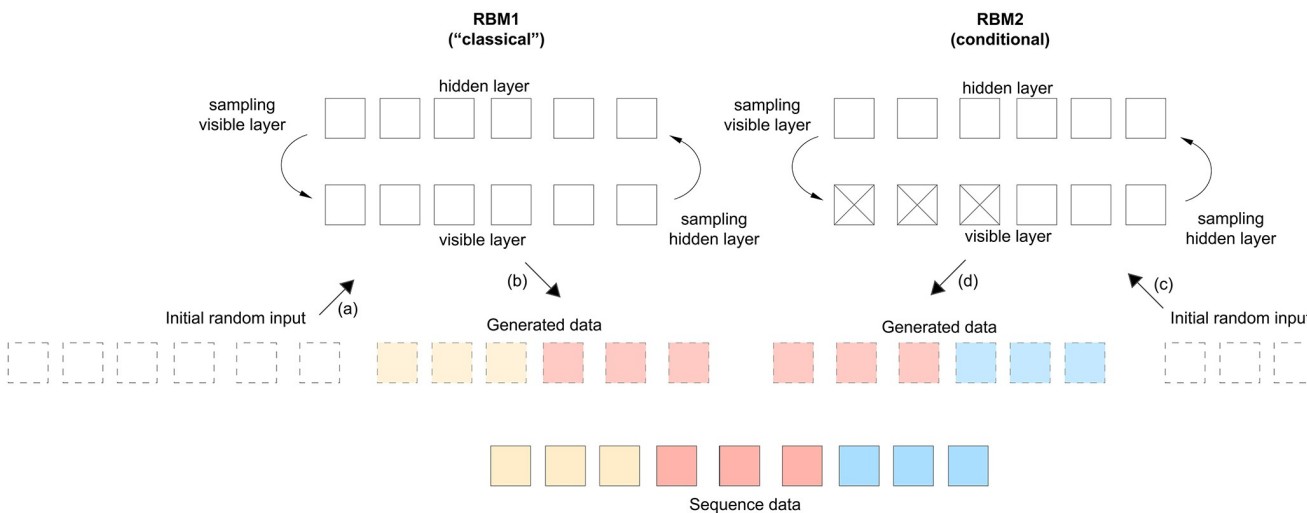

**Fig 2. Illustration of the learning and sampling of a large sequence using a "classical" and a conditional RBM (CRBM).** Initially, we train RBM1 (left) and RBM2 (right) in parallel. Both RBMs are essentially trained in a similar manner: random inputs are drawn and k MC steps are performed before computing the gradient and updating the weights using gradient descent. The difference for the CRBM (RBM2) is that half of the variables in the visible layer are pinned (crossed squares) to the real data during training while the rest is generated conditionally on these pinned variables. After training both machines, we can sample a complete new sequence. To do so, we start from random input and perform k MC steps to generate the first part of the sequence (light yellow-red) using RBM1. Then, we use half of this generated sequence (light red) as the pinned visible variables of the RBM2 (crossed squares) and initialise the rest as random input. We perform k MC steps on RBM2 while keeping the pinned variables fixed to generate the rest of the sequence (light blue). The letters next to arrows show the order of this sampling procedure.

For the generation of very long sequences, we designed a novel procedure based on conditional-RBMs (CRBMs) [23, 35] (Fig 2). The CRBM consists in the learning of correlation patterns conditionally to some input variables. Hence, instead of considering all the dataset $X$, we separate it into two parts $X = X_P \cup X_I$, and the CRBM will learn to generate $X_I$ (the inferred variables) based on $X_P$ (the pinned variables). Therefore, we design a gradient that will learn how to generate the variables $X_I$ given that we provide the variables $X_P$. In practice, denoting $\underline{s}$ as the variables to be inferred and $\underline{x}$ as the pinned ones, we will maximize the following quantity:

$$p(\underline{s} \in I | \underline{x} \in P) = \frac{1}{Z} \exp\left(\sum_{ia} w_{ia} s_i \tau_a + \sum_i \theta_i s_i + \sum_a \eta_a \tau_a + \sum_a \tau_a \sum_j w_{ja} x_j\right)$$

Therefore, for each sample of the dataset, this construction is similar to a classical RBM with additional biases for the hidden nodes which depend on the pinned variables. During the learning, we infer the parameters of the model such that when showing a configuration of pinned variables $X_P$, the model will generate a sample that is correlated to it accordingly. This conditional model can be used to learn very long chains of variables. Let us consider that we have a dataset with 100,000 input variables. We can learn initially a regular RBM on the first 10,000 input variables. In parallel, we can also learn a conditional RBM on the input variables $s_i$ with $i$ in [5,000:15,000] using the first 5,000 variables as pinned variables. Therefore, this second RBM will, given 5,000 pinned variables, learn how to generate the 5,000 following ones. The same procedure is repeated with various input sets, always using the first 5,000 input nodes as pinned variables. Once learning is completed for all the RBMs, we can proceed using the following sequential method to generate new data:

1. Use the first RBM (non-conditional) to generate the first 10,000 input.

2. Use the first CRBM to generate the next [10,000:15,000] inputs. To do that, we use the generated nodes [5,000:10,000] as pinned variables and generate the rest.

3. We follow the same procedure until we finally generate the whole 100,000 variables.

This method was used to generate the large-scale 65,535-SNP dataset. We first trained a "normal" RBM on the first 5,000 input variables. Then, we made a set of 10,000-input conditional RBMs, where the first 5,000 variables were used as pinned variables. All the models were trained with 1000 hidden nodes, and a learning rate of $\beta = 0.005$. The learning dynamics uses the Rdm-k method: each chain was generated starting from the uniform Bernoulli distribution, with k = 100 for the training of the RBM and the CRBMs. A recurrent issue with RBM (or training in general) is to decide when to stop the training. While in supervised setting it is easy to monitor the loss function (or the number of correctly classified samples), it is not the case for RBM since the partition function is intractable. In this work, the learning was affected by the small number of samples given the dimension of the inputs. Therefore (and to avoid overfitting), the meta-parameters such as the number of hidden nodes and the number of epochs were fixed a posteriori, by investigating various trained machines with different values of these parameters and choosing the one giving a good $AA_{TS}$ score. The detailed python scripts can be accessed at https://gitlab.inria.fr/ml_genetics/public/artificial_genomes. The RBM implementations are based on the pytorch library and handle GPU to perform both training and sampling. Another example of the out-of-equilibrium code applied to images can be found at https://github.com/AurelienDecelle/TorchRBM.

## VAE implementation

The VAE [36] architecture is very similar to the GAN architecture but somehow reversed: the encoder is an analogue to the critic and the decoder is an analogue to the generator (S1 Fig). The last layer of the encoder outputs two vectors containing means ($\mu$) and standard deviations ($\sigma$) so that the latent space can be sampled based on these values and fed to the decoder. The loss function consists of the reconstruction term, which measures the likelihood of the generated genomes (via log loss in our implementation), and the regularisation term which is the Kullback-Leibler divergence between the standard normal distribution and the prior distribution of the latent space. The regularisation term directs the latent space towards a standard normal distribution. After training completion, this allows sampling of new latent points from the standard distribution which are further transformed into new data points (new genomic sequences) through the decoder.

Each layer in our implementation is followed by batch normalisation and leaky ReLU with alpha set to 0.01, except for the final layers. The decoder final layer is followed by a sigmoid function, whereas the encoder final $\mu$ and $\sigma$ layers have linear activation. We used Adam optimizer with default settings and the learning rate set to 0.001. Similarly to the WGAN, we evaluated the training based on coherence of the PCA performed on real and generated genomes (components 1 to 4). We could not successfully train a VAE model which generates plausible AGs for the 65,535-SNP dataset. Further architecture and hyperparameter optimization is needed to better assess VAE models in this context. VAE models were coded with python-3.9 and pythorch-1.11 [32]. The detailed python scripts can be accessed at https://gitlab.inria.fr/ml_genetics/public/artificial_genomes.

## Nearest neighbour adversarial accuracy ($AA_{TS}$)

Similarly to [21], we assessed the overfitting/underfitting characteristics of AGs using the $AA_{TS}$ score [37]. $AA_{TS}$ is calculated as follows:

$$AA_{truth} = \frac{1}{n} \sum_{i=1}^{n} \mathbf{1}(d_{TS}(i) > d_{TT}(i))$$

$$AA_{syn} = \frac{1}{n} \sum_{i=1}^{n} \mathbf{1}(d_{ST}(i) > d_{SS}(i))$$

$$AA_{TS} = \frac{1}{2}(AA_{truth} + AA_{syn})$$

where $n$ is the number of samples in each dataset (real and generated), $d_{TS}(i)$ is the distance between the real (truth—T) sample indexed by $i$ and its nearest neighbour in the generated (synthetic—S) dataset, $d_{TT}(i)$ is the distance between a real sample $i$ and its nearest neighbour in the real dataset, $d_{ST}(i)$ is the distance between a generated sample $i$ and its nearest neighbour in the real dataset, $d_{SS}(i)$ is the distance between a generated sample $i$ and its nearest neighbour in the generated dataset, and $\mathbf{1}$ is the indicator function which returns 1 if the argument is true and 0 otherwise. Based on this equation, an $AA_{TS}$ value below 0.5 indicates overfitting and an $AA_{TS}$ value above 0.5 indicates underfitting. We additionally obtained a privacy score in another analysis where we separated the datasets into two equal-sized train and test sets (2504 phased haplotypes each). The score is defined as follows:

$$Privacy\ score = Test\ AA_{TS} - Train\ AA_{TS}$$

where $Test\ AA_{TS}$ (resp. $Train\ AA_{TS}$) is the $AA_{TS}$ computed with the test (resp. training) samples as truth. The expected value for the privacy score is 0 when there is no privacy leakage, with higher scores indicating higher leaks.

## Summary statistics

For the 65,535-SNP dataset, LD was computed only on a subset of pairs in order to fasten computation. To sample these pairs in an efficient way (i.e. approximately uniformly along the SNP distance log scale without computing the full matrix of SNP distances), we used the script from [38]. The remaining summary statistics (allele frequencies, haplotypic pairwise distances, PC scores, 3-point correlations and LD for the 10,000-SNP dataset) were computed as in [21] using the publicly available scripts at https://gitlab.inria.fr/ml_genetics/public/artificial_genomes. For the radar plots, we transformed the scores so that they span values between 0 and 1, where 0 represents poor performance and 1 high or perfect performance. Precisely, we used the allele frequency correlation for alleles with low frequency ($<= 0.2$) for the *Allele frequency* score and correlation of SNPs separated by random distances for the *3–point correlations* score. For the *Pairwise distance* score, we used the Wasserstein distance between the distributions of haplotypic pairwise distances of real and generated data. We performed min-max scaling, using the value for a simple binomial generator model (from [21]) for the lowest

bound 0. For the other scores, we used the following equations:

$$Overfitting = 1 - (0.5 - \min(AA_{truth}, 0.5)) - (0.5 - \min(AA_{syn}, 0.5))$$

$$Underfitting = 1 - (\max(AA_{truth}, 0.5) - 0.5) - (\max(AA_{syn}, 0.5) - 0.5)$$

$$LD = 1 - \sum(LD_{real} - LD_{generated})^2 \Big/ \sum LD_{real}^2$$

where $LD_{generated}$ and $LD_{real}$ are average LD values for bins in the LD decay analyses. *Overfitting* and *Underfitting* equations were formulated to focus on below and above 0.5 $AA_{TS}$ sweet spot, respectively, to provide a better resolution in terms of overfitting/underfitting assessment.

To analyze the correlation of k-mer haplotypes between real and generated genomes, we divided 10,000-SNP and 65,535-SNP genomes into non-overlapping windows of size 4 and 8. We counted the number of occurrences of each unique motif in these windows and assessed the correlation of these counts between real and generated genomes for each window.

## Nearest neighbour chain analysis

Since $AA_{TS}$ scores for the CRBM AGs were anomalous, we perfomed a nearest neighbour chain analysis for further investigation. For this analysis, the frequencies of all observable patterns for the nearest neighbour chains of size 2 to 5 were computed. A pattern indicates the succession of data type (synthetic/S or truth/T) when starting from a point (the first letter) and moving successively to the nearest neighbour, then the nearest neighbour of the nearest neighbour, and so on until reaching the chain size. Hamming distance was used for identifying the nearest neighbours.

## Membership inference attacks

We performed membership inference attacks on WGAN and RBM generated AGs using the approach proposed in [39]. We trained the WGAN and RBM models using half of the haplotypes (2504) of the 10,000-SNP data and kept the rest as test set. We considered the two following scenarios: a white-box attack where the adversary has access to the original critic optimized weights and architecture, and a black-box attack where the adversary has only access to the WGAN architecture (without its weights) and generated samples. For both scenarios, we also assume the adversary knows the size of the original training set and has a collection of samples some of which are suspected to belong to the training set. For the white-box attack, we used the already trained critic to score all the samples (a total of 5008 samples consisting of 2504 from training and 2504 from test sets) and sorted them based on these scores. The top $n$ ranking samples are then predicted as belonging to the training set. For the black-box attack, we trained new models on generated AGs, using the exact same architecture as the previously trained WGAN, and the same stopping criterion, but we overtrained further up to 5000 epochs to induce overfitting. Since this attack is model agnostic, we trained one model on WGAN AGs and one on RBM AGs. Similar to the white-box attack, all samples are scored by the critic after the training and the top $n$ ranking samples are assigned to the training set. In our experiments, $n$ varies such that we assign 1%, 10%, 25%, or 50% of the total 5008 samples to the training set. The computed accuracy is simply the percentage of correct assignments for each value of $n$.

## Results

### Comparisons with the previous models

As our GAN concept has changed substantially compared to the previous models both in terms of architecture and loss functions, we initially performed training and analysis on 1000 Genomes data with the same 10,000 SNPs as [21] to be able to conduct one-to-one comparisons. For these tests, we additionally implemented a new RBM scheme along with a new variational autoencoder (VAE) which has a very similar architecture to our WGAN model. We used the VAE as a supplementary benchmark since the encoder-decoder form of the VAE can be seen as an analogue to the critic-generator form of the WGAN model. The objective function of VAE, on the other hand, is substantially different, which allows us to assess the robustness of the architecture we used (see Materials and methods).

Based on the PCAs, all models generated AGs which could capture the population structure of the data, albeit new WGAN and RBM models were better at representing the real PC densities compared to the other models (Fig 3A). In our previous study, we reported that the GAN model had difficulty learning low frequency alleles observed in real genomes. The new WGAN and RBM showed improvement in capturing the real allele frequency distribution, especially for the low frequency ($<= 0.2$) alleles with correlation coefficients for previous GAN (GAN_prev), previous RBM (RBM_prev), VAE, WGAN and RBM being 0.94, 0.83, 0.94, 0.96 and 0.99, respectively (Fig 3B). In terms of LD structure, RBM generated AGs seemed to have the closest decay scheme to the real data while WGAN, GAN and VAE generated AGs all had similar results without substantial difference (S2 Fig). In addition, the new WGAN and RBM models preserved 3-point correlations better than the other models (S3 Fig). To assess whether AGs could preserve short motifs in real genomes, we also analyzed the correlation of non-overlapping 4-mer and 8-mer motifs between real and generated datasets (S4 Fig). AGs generated by all models showed high correlation and good fit overall, although RBM_prev had the highest variance despite the high correlation coefficient.

None of the models have produced identical sequences and no full sequences were copied from the training dataset except for the benchmark VAE. In addition, the distribution of haplotypic pairwise differences between real and generated datasets overlapped well with the distribution of haplotypic pairwise differences within the real dataset (S5 Fig). The $AA_{TS}$ scores for GAN_prev, RBM_prev, VAE, WGAN and RBM were 0.73 ($AA_{truth}$ = 0.57, $AA_{syn}$ = 0.89), 0.49 ($AA_{truth}$ = 0.46, $AA_{syn}$ = 0.52), 0.48 ($AA_{truth}$ = 0.47, $AA_{syn}$ = 0.50), 0.82 ($AA_{truth}$ = 0.77, $AA_{syn}$ = 0.87) and 0.47 ($AA_{truth}$ = 0.47, $AA_{syn}$ = 0.47), respectively. These values indicate underfitting (a hypothetical extreme case demonstrated in Fig 4A) for GAN generated AGs, and slight overfitting (a hypothetical extreme case demonstrated in Fig 4B) for the RBM and VAE generated AGs (S6 Fig). Although the new WGAN demonstrated slightly more underfitting than the previous GAN, the gap between the two components of $AA_{TS}$ ($AA_{truth}$ and $AA_{syn}$) was decreased (see Materials and methods for the details of the terminology). We previously hypothesised that the low value of $AA_{truth}$ and high value of $AA_{syn}$ for the previous GAN might be due to the generator creating AGs based on averages from a local set of samples in small pockets (extreme case demonstrated in Fig 4C). This can be seen as a generative aberration which does not generalise to the whole dataset but is observed only regionally in small subsections of the data. We do not observe this behaviour in the new WGAN model. A general comparison of all the models based on multiple aggregated statistics is provided in S7 Fig.

### Generating large-scale genomic data

Following the main motivation of the study, we trained the new WGAN and CRBM models on 1000 Genomes data with 65,535 SNPs. The WGAN model for this data was deeper

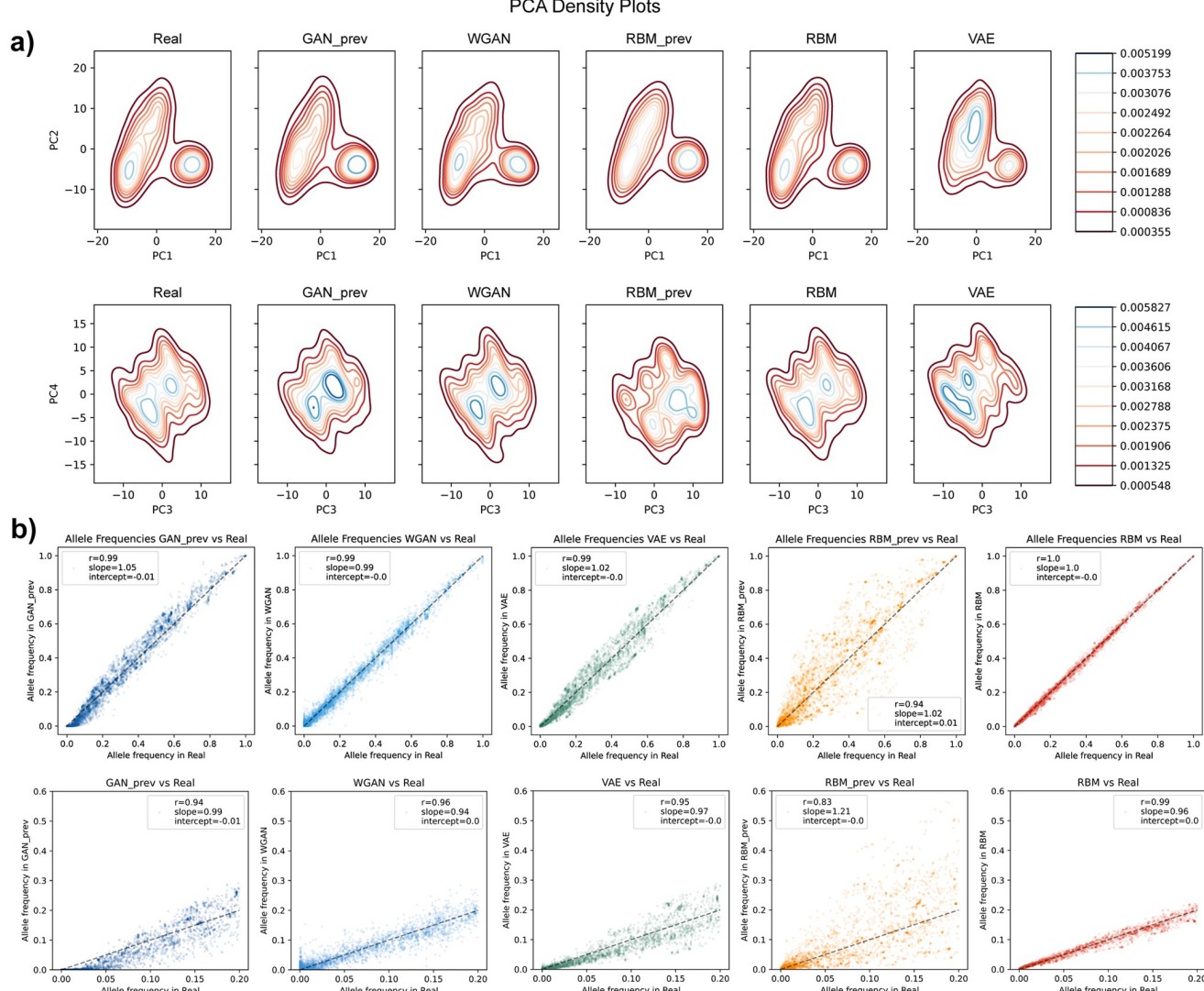

**Fig 3. Principal component and allele frequency analyses of artificial genomes with 10,000-SNP size. a)** Density plot of the PCA of combined real and artificial genome datasets. Density increases from red to blue. (**b**) Allele frequency correlation between real (x-axis) and artificial (y-axis) genome datasets. Bottom figures are zoomed at low frequency alleles (from 0 to 0.2 overall frequency in the real dataset). Values presented inside the figures are Pearson's r, ordinary least squares regression slope and intercept.

compared to the model used for the 10,000-SNP data (see Materials and methods). We again implemented a VAE similar to the WGAN architecture yet we could not train this model with satisfactory results (see Discussion). Both WGAN and CRBM generated AGs were able to capture the real population structure and PCA modes quite well (Fig 5A) along with allele frequencies (correlation coefficient for low frequency alleles being 0.97 for both models; Fig 5B), yet WGAN AGs had substantially more fixed alleles which had low frequency in the real dataset (S8 Fig). Since computation of the full correlation matrix is very intensive due to large sequence size, we calculated an approximation of the LD decay based on a subset of SNP pairs (see Materials and methods). AGs generated by both models had on average lower LD than real genomes, similarly to our previous findings (Fig 5C). In 3-point correlation analysis,

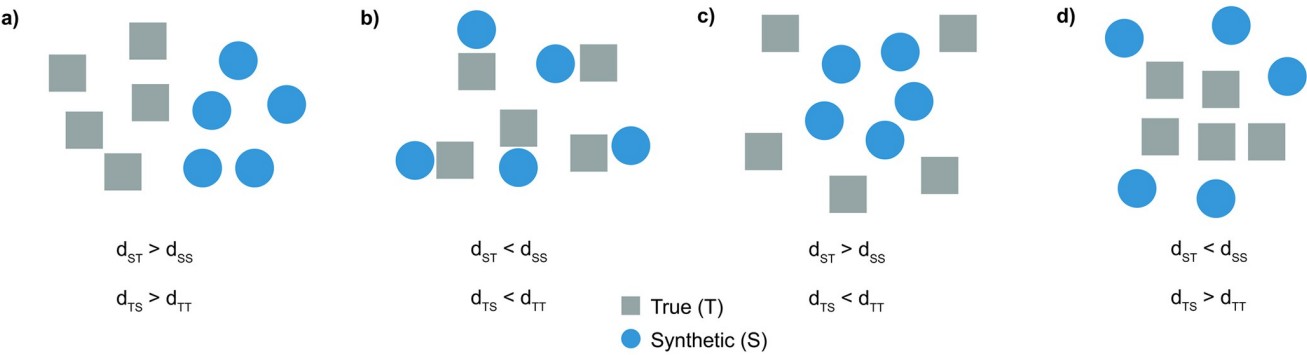

**Fig 4. Schematic representation of different problematic training outcomes for generative models.** Distances to the nearest neighbours are denoted by $d_{xx}$ where $x$ can be T (truth/real) and S (synthetic/generated). **a)** An extreme case of underfitting (or optimization issue) in which the nearest neighbours of real data points are real and the nearest neighbours of synthetic data points are synthetic, revealing two distinct clusters ($AA_{TS} \gg 0.5$). **b)** An extreme case of overfitting in which the nearest neighbours of real data points are systematically synthetic and vice versa ($AA_{TS} \ll 0.5$). **c)** An extreme case of a specific type of generative aberration in which the nearest neighbours of both real and synthetic data points are synthetic ($AA_{syn} \gg 0.5$ and $AA_{truth} \ll 0.5$). Hypothetically, this might occur when the generator generates new instances based on average information from a small collection of samples, causing low local variation. **d)** An extreme case of a specific type of generative aberration in which the nearest neighbours of both real and synthetic data points are real ($AA_{syn} \ll 0.5$ and $AA_{truth} \gg 0.5$). This might possibly be observed when the generative model learns the main modes in real data but fails to mimic the densities and generates instances in the axes of the main modes with high variance.

CRBM performed better than WGAN for SNP triplets seperated by 1, 4, 16, 64, 256, 512 and 1024 SNPs. However, the score was similar for SNP triplets seperated by random distances (WGAN = 0.43, CRBM = 0.41; S9 Fig). Furthermore, 4-mer and 8-mer motif distributions both for WGAN and CRBM generated AGs seemed to be similar to real genomes (S10 Fig).

Similarly to the 10,000-SNP dataset, neither of the models produced identical sequences and no full sequences were copied from the training dataset. WGAN captured the haplotypic pairwise distribution better than CRBM (pairwise distance radar score for WGAN = 1.00, CRBM = 0.94; S11 Fig) but the two main peaks in real data were correctly represented by both models (S12 Fig)). The $AA_{TS}$ value for WGAN AGs showed underfitting ($AA_{TS} = 0.91$) whereas the value for CRBM was slightly above the 0.5 sweet spot ($AA_{TS} = 0.56$; Fig 5D. However, there was a huge contrast between $AA_{truth}$ and $AA_{syn}$ values for CRBM AGs ($AA_{truth} = 0.86$, $AA_{syn} = 0.26$), which might be an indication of the anomaly depicted in Fig 4D. This is the opposite of what we observed for the previous GAN (GAN_prev) model and might be due to AGs which exist outside the real data space as can be seen from the PCA analysis (S13 Fig). A general comparison of the two models based on multiple aggregated statistics is provided in S11 Fig.

To further investigate the anomaly for the CRBM AGs, we performed a nearest neighbour chain analysis (Table 1). Starting with a synthetic point, we observed substantially higher frequencies for chains of true data points (ST, STT, STTT, STTTT) compared to chains of synthetic data points (SS, SSS, SSSS, SSSSS).

## Membership inference attacks and privacy leakage

Assessing privacy leakage and performing membership inference attacks require an additional test set but we could not obtain good quality AGs using a subset of 65,535-SNP dataset. Therefore, we trained the new WGAN and RBM models using half of the samples from the 10,000-SNP dataset (2504 haplotypes). AGs generated via the WGAN model had similar summary statistics to the AGs generated via the model trained with the whole dataset. For the RBM model, results were slightly worse but better than the previous RBM model (RBM_prev)

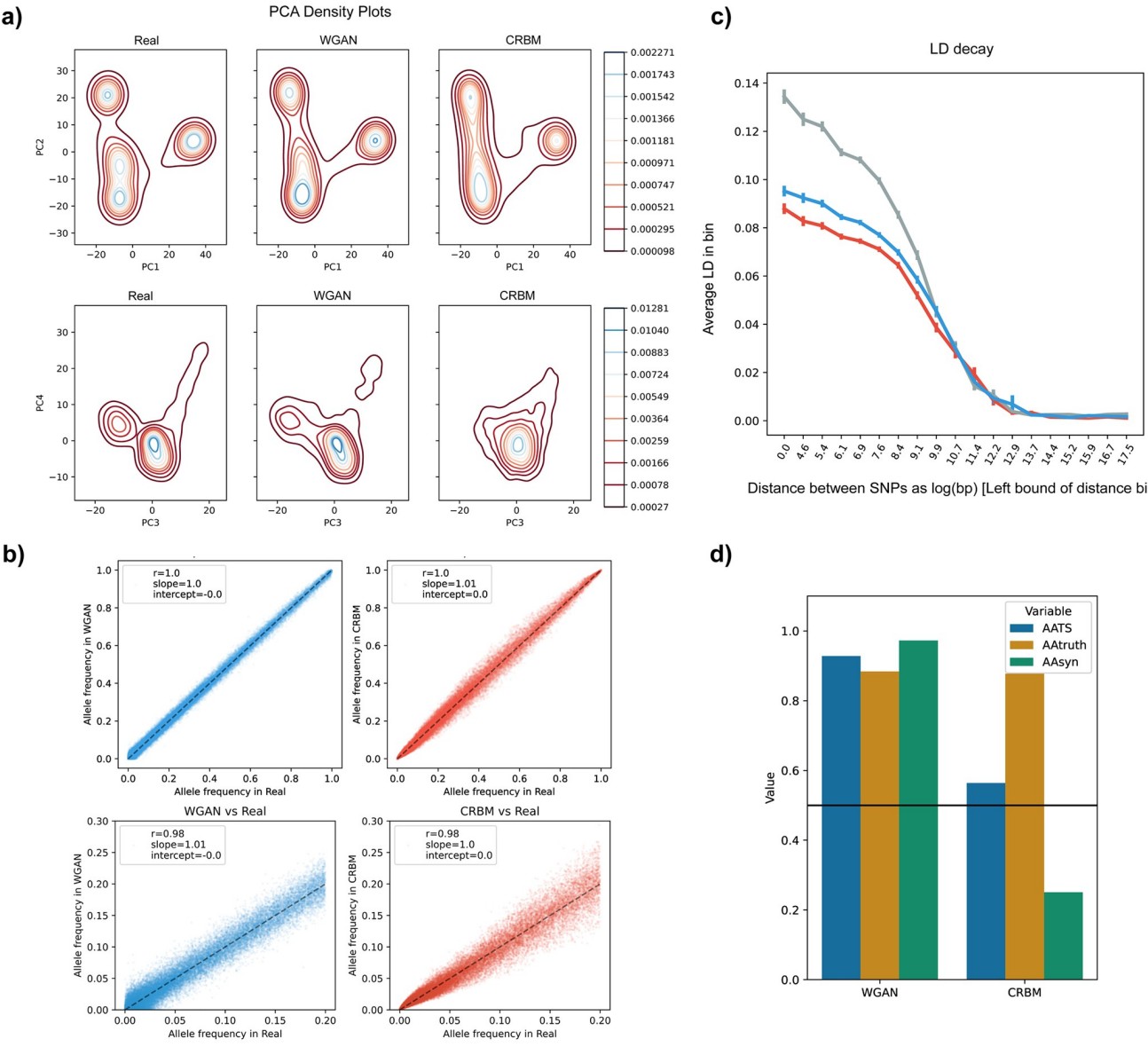

**Fig 5. Principal component, allele frequency and linkage disequilibrium (LD) analyses of artificial genomes with 65,535-SNP size. a)** Density plot of the PCA of combined real genomes and artificial genomes generated by WGAN and CRBM. Density increases from red to blue. **b)** Allele frequency correlation between real and artificial genome datasets. Bottom figures are zoomed at low frequency alleles (from 0 to 0.2 overall frequency in the real dataset). Values presented inside the figures are Pearson's r, ordinary least squares regression slope and intercept. The dashed black line is the identity line. **c)** LD decay approximation for real (grey), WGAN generated (blue) and CRBM generated (red) genomes (see Materials and methods for details). **d)** Nearest neighbour adversarial accuracy (AATS) of artificial genomes generated by different models for the 65,535-SNP dataset. Values below 0.5 (black line) indicate overfitting and values above indicate underfitting.

trained with the whole dataset (S14A–S14C Fig). $AA_{TS}$ analysis showed underfitting for WGAN AGs ($AA_{TS}$ = 0.80, $AA_{truth}$ = 0.76, $AA_{syn}$ = 0.83) and almost perfect scores for RBM AGs ($AA_{TS}$ = 0.50, $AA_{truth}$ = 0.49, $AA_{syn}$ = 0.51). Based on the test set, WGAN AGs had good privacy score (0.03) and RBM AGs showed possible privacy leakage (0.23) (S14D Fig). Further-more, both white-box attack on WGAN AGs and black-box attack on RBM AGs reached high accuracies (WGAN: 1%: 0.82, 10%: 0.826, 25%: 0.754, 50%: 0.677; RBM: 1%: 0.84, 10%: 0.70,

**Table 1. Nearest neighbour chain analysis.** Frequencies of series of generated (synthetic—S) and real (truth—T) samples in chains of nearest neighbours (from size 2 to 5). To avoid loops, a sample was removed once reached in the chain. Expected frequency for chains of size 2 is 0.25, size 3 is 0.125, size 4 is 0.0625 and size 5 is 0.03125.

| SS | ST | SSS | SST | STT |
|---|---|---|---|---|
| 0.1258 | 0.3742 | 0.0519 | 0.0739 | 0.3267 |
| **TT** | **TS** | **TTT** | **TTS** | **TSS** |
| 0.4389 | 0.0611 | 0.3965 | 0.0424 | 0.0167 |
| **STS** | **SSSS** | **STTT** | **SSSSS** | **STTTT** |
| 0.0475 | 0.0240 | 0.2891 | 0.0110 | 0.2575 |
| **TST** | **TTTT** | **TSSS** | **TTTTT** | **TSSSS** |
| 0.0444 | 0.3593 | 0.0061 | 0.3270 | 0.0017 |

25%: 0.62, 50%: 0.57) for detecting a portion of the training samples whereas black-box attack on WGAN AGs produced substantially lower accuracies (1%: 0.56, 10%: 0.55, 25%: 0.54, 50%: 0.51), suggesting relatively better privacy preservation. The distribution of the critic scores for train and test samples also showed that the black-box attack on WGAN AGs was mainly unsuccessful for differentiating training and test samples (S15 Fig).

## Discussion

In this study, we implemented generative neural networks for large-scale genomic data generation and assessed various characteristics of the artificial genomes (AGs) generated by these networks. Initially, we generated AGs from the 10,000-SNP dataset to be able to compare to the previous results from [21]. For this dataset, we introduced a new RBM training scheme, a convolutional WGAN and a convolutional VAE. Both WGAN and new RBM models substantially improved the quality of AGs in terms of all summary statistics. For the 65,535-SNP dataset, we used a similar WGAN architecture (which is essentially deeper in comparison to the architecture for the 10,000-SNP dataset) and a conditional RBM (CRBM) protocol. AGs generated by these new models preserve population structure, allele frequency distribution and haplotypic integrity of real genomes reasonably well with little or no privacy leakage from the training data. Generative models trained with similar-sized genomic data have been reported in the literature but the main goal of these studies was characterization of population structure via dimensionality reduction and the generated genomes did not possess good haplotypic integrity [27, 40]. There have been other studies focusing on demographic parameter estimation [41] and data generation [8–11] for population genetics but these only included training with smaller genomic segments.

The upscaling to a larger sequence size in comparison to our previous study is an essential step for AGs to be utilised in real-life applications as publicly accessible alternatives for sensitive genomic samples, yet obstacles remain. Although we could generate substantially larger sequences by incorporating convolutions for the WGAN and a conditional approach for the RBM, training these large models requires computational time and fine-tuning in most cases. The WGAN is usually preferred over naive GAN in literature as it mitigates hyperparameter optimization and provide more stable training [22]. However, we had to go through multiple combinations of architectures and hyperparameter values to reach satisfying results. Even then, the training in our application involved further adjustment of the generator learning rate after a certain epoch to capture the details of the population structure observable in higher PC dimensions. This second round of training was helpful but some of the runs did not reach an acceptable equilibrium (i.e., generated genomes had low quality after training), especially

when experimenting with reduced training sample sizes. For future applications, automated architecture and hyperparameter tuning methods can help significantly for finding the optimal combinations without extensive trial and error [42, 43]. VAEs might be seen as better alternatives in terms of training stability compared to GANs since they do not suffer from the difficulty of balancing two networks in an adversarial manner, yet we could not obtain high-quality AGs for the 65,535-SNP dataset. Presumably, considerably better outcomes can be achieved with further architectural, hyperparameter and loss function exploration, which is outside the scope of this study.

Another future research direction for the generative models could be the exploration of alternative stopping criteria for the training. Unlike image generation, where the generated outcomes can be assessed easily by visual inspection, assessment of generated genomic data is not trivial. For VAE and WGAN models, we decided to use PCA plots for initial inspection and as the stopping criterion since the highest variation in most genomic datasets is due to population structure which PCA captures well. This PCA match, inspected visually or measured through Wassertein distance, had been used previously as the stopping criterion [8, 21]. Although an obvious candidate for the stopping criterion of WGAN is the convergence of the critic's loss to a value close to zero (since this loss provides an estimation for the Earth mover's distance between real and generated data), we observed that it does not always coincide with good PCA outcomes for the AGs. For the RBM models, we had to inspect $AA_{TS}$ score instead since overfitting was a more prominent issue. A possible alternative to these would be using some aggregate statistics crafted based on multiple summary statistics related to LD, site frequency spectrum, ancestry and overfitting scores.

The other approach we presented to incorporate large-scale data was the conditional training of RBMs. Instead of a single training of a large and deep model as in the case of WGAN, CRBM training includes multiple training runs of a small model. A main advantage of this method is that any sequence size can theoretically be generated as long as the positional conditionality is not broken over the target genome segment. Therefore, the only bottleneck in terms of computation is the time needed to train all the RBMs. Currently, the learned weights are specific to each machine and no parameters are shared between RBMs. This advantageously allows parallel training when needed, but produces a large number of parameters to be optimized since the number of weights and biases has to be multiplied by the number of machines. Further studies could investigate whether parameter sharing between machines (i.e., between regions) and/or within each machine (through convolutions) is advantageous or instead complicates training. In addition, many datasets also present a difficulty during training, because the equilibrating time for Markov chains diverges as the training goes on, causing issues for sampling (and re-sampling) [44]. To overcome this, we relied on out-of-equilibrium training as described in Materials and Methods. While this approach improved training stability and also the quality of the generated samples with respect to other approaches, it has the disadvantage that the learned features cannot be easily interpreted.

For the assessment of overfitting and underfitting, we utilised the $AA_{TS}$ score and haplotypic pairwise distances as in [21]. An interesting finding was the large difference between two terms of the $AA_{TS}$ score ($AA_{truth}$ and $AA_{syn}$) for AGs generated by the CRBM even though the averaged $AA_{TS}$ score was good. Interestingly, this is the opposite of what we had observed for our previously published GAN model and possibly points to the type of generative aberration demonstrated in Fig 4D. In fact, it is known that the likelihood function used to infer the parameters of RBMs tends to create potentially spurious modes -that do not match any region of the dataset-, which might explain these patterns. For further investigation, we performed a nearest neighbour chain analysis since we would expect the nearest neighbours of the nearest neighbours for CRBM AGs to be mostly real genomes with this anomaly (Table 1). High

frequency of ST, STT, STTT, STTTT and low frequency of SS, SSS, SSSS, SSSSS measurements provide additional evidence that what we observe for the CRBM AGs is possibly similar to the scenario seen in Fig 4D. To the best of our knowledge, this type of anomaly for generative models has not been reported in literature and could have been missed by classical evaluation metrics. Such complex overfitting/underfitting phenomena require vigilance and further investigation in future generative studies.

We furthermore performed membership inference attacks on AGs generated by WGAN and RBM models to assess possible privacy leakage. While the white-box attack on WGAN AGs and black-box attack on RBM AGs produced high accuracy for detecting a portion of the samples used in training, black-box attack on WGAN AGs was mainly not successful (S15 Fig). This indicates that even if the model architecture is available publicly and the adversary is in possession of some samples from the training dataset, it is not trivial to pinpoint the training samples via this attack without access to the model weights. We highlight that providing the critic's weights of a GAN is not useful to the general user interested only in its generative properties, the white-box is thus a conservative attack (i.e., unrealistically beneficial for the adversary). Similarly, both the white and black-box attacks are conservative for evaluating privacy leaks as they assume that the adversary has the huge advantage of already possessing some genetic sequences from the original training set. However, it is crucial to underline here that this is only a single type of attack and more research on privacy preservation is essential before AGs are used in real-life scenarios [45].

Despite these issues which remain to be further studied, additional improvements in model training and increased privacy guarantees for generated genomes can pave the way for the first artificial genome banks in the near future, accelerating global access to the vast amount of restricted genomic data. Unlike conventional approaches for genomic simulations, generative neural networks do not require a priori information about the target dataset (such as underlying evolutionary history), allow the generated data to be utilized alongside real data and provide substantially better privacy outcomes in comparison to haplotype copying methods [21].

## Supporting information

**S1 Fig. Architecture of the variational autoencoder (VAE) model for the 10,000 SNP dataset.** Generic blocks of the encoder and decoder (white rectangles) are conceptually the same with the generic critic and generator blocks respectively (Fig 1A), except that there are no latent space channels concatenated to the input and no additional noise vectors at each block. The major difference from WGAN in terms of architecture is the last block of the encoder, which encodes mu and sigma as the mean and the standard deviation of the generated distribution, which are used to sample the latent space. Dotted connections are residual connections where the input value is added to the output value of the block before passing to the next block. Numbers in parentheses above blocks show channels and length, respectively (C, L). (TIF)

**S2 Fig. Comparative linkage disequilibrium (LD) analysis of artificial genomes generated by different models for the 10,000-SNP data. a)** LD heatmap based on $r^2$ matrices. Sections below diagonals correspond to LD in real genomes and sections above diagonals correspond to LD in artificial genomes. **b)** LD decay as a function of SNP distance. SNPs were binned based on distance and average LD was calculated. **c)** LD decay correlation between real and artificial datasets. x axis corresponds to real LD bins and y axis corresponds to generated LD bins. Sites fixed in any of the datasets were removed for all the LD calculations. (TIF)

**S3 Fig. 3-point correlation analysis of SNP triplets for the 10,000-SNP data with inter-SNP distances of 1, 4, 16, 64, 256, 512 and 1024 (from left to right, top to bottom).** The last panel (bottom right) shows correlation for triplets of SNPs drawn randomly. In each plot, drawing order (z-order) of each AG group is shuffled.
(TIF)

**S4 Fig. Analysis of small haplotype motifs between real and generated 10,000-SNP datasets for a) 4-mer and b) 8-mer non-overlapping windows.** For each unique k-mer in each window, number of occurrences in the real dataset was compared to the same number in the AG dataset. Each point corresponds to the occurrence number in real (x-axis) and AG (y-axis) datasets. Values presented inside the figures are Pearson's r, ordinary least squares regression slope and intercept.
(TIF)

**S5 Fig. Distribution of haplotypic pairwise difference within (left figure) and between (right figure) 10,000-SNP datasets.**
(TIF)

**S6 Fig. Nearest neighbour adversarial accuracy (AATS) of artificial genomes generated by different models for the 10,000-SNP dataset.** Values below 0.5 (black line) indicate overfitting and values above indicate underfitting. See Materials and methods for the details of the metrics.
(TIF)

**S7 Fig. Radar plot comparing artificial genomes generated by different models for the 10,000-SNP dataset.** Values closer to 0 indicate poor performance whereas values closer to 1 indicate good performance. See Materials and methods for the details of the representative statistics.
(TIF)

**S8 Fig. Analysis of fixed alleles in artificial genomes with 65,535 SNPs.** Left figures show the number of fixed alleles in artificial genomes (x axis) versus the frequency of these alleles in the real dataset (y axis). Right figures show the distribution of the frequency of alleles fixed in the artificial dataset but not fixed in the real dataset.
(TIF)

**S9 Fig. 3-point correlation analysis of SNP triplets for the 65,535-SNP data with inter-SNP distances of 1, 4, 16, 64, 256, 512 and 1024 (from left to right, top to bottom) for a) WGAN and b) CRBM AGs.** The last panels (bottom right) shows correlation for triplets of SNPs drawn randomly. In each plot, drawing order (z-order) of each AG group is shuffled.
(TIF)

**S10 Fig. Analysis of small haplotype motifs between real and generated 65,535-SNP datasets for a) 4-mer and b) 8-mer non-overlapping windows.** For each unique k-mer in each window, number of occurrences in the real dataset was compared to the same number in the AG dataset. Each point corresponds to the occurrence number in real (x-axis) and AG (y-axis) datasets. Values presented inside the figures are Pearson's r, ordinary least squares regression slope and intercept.
(TIF)

**S11 Fig. Radar plot comparing artificial genomes generated by WGAN and CRBM models for the 65,535-SNP dataset.** Values closer to 0 indicate poor performance whereas values closer to 1 indicate good performance. See Materials and methods for the details of the

representative statistics.
(TIF)

**S12 Fig. Distribution of haplotypic pairwise difference within (left figure) and between (right figure) 65,535-SNP datasets.**
(TIF)

**S13 Fig. Principal component analysis (PCA) of combined real and artificial genomes with 65,535 SNPs.**
(TIF)

**S14 Fig. Analysis of WGAN and RBM generated AGs using 2504 samples from 10,000-SNP dataset. a)** Principal component analysis (PCA) of combined real and artificial genomes. **b)** Allele frequency correlation between real (x-axis) and artificial (y-axis) genome datasets. Bottom figures are zoomed at low frequency alleles (from 0 to 0.2 overall frequency in the real dataset). Values presented inside the figures are Pearson's r, ordinary least squares regression slope and intercept. **c)** Nearest neighbour adversarial accuracy (AATS) of artificial genomes generated by different models and the test set. Values below 0.5 (black line) indicate overfitting and values above indicate underfitting. **d)** Privacy score for WGAN and RBM generated AGs. Values close to 0 indicate no privacy leakage.
(TIF)

**S15 Fig. Membership inference attack on generated 10,000-SNP genomes. a)** White-box attack (adversary has access to the model architecture and weights) on WGAN AGs and black-box attacks with auxiliary information (adversary has only access to the model architecture) on **b)** WGAN and **c)** RBM AGs. For all attacks, the adversary is assumed to know the size of the training set (2504 in this analysis) and possesses a set of samples (5008 in this analysis) suspected of belonging to the training data. For each attack, the critic scores the samples and the adversary sets a threshold for assigning the top $n$ scoring samples to the training dataset. Figures in the upper row show the accuracy of attacks depending on these thresholds (assigned samples ranging from the top 1% to the top 50%). The red dashed lines indicate the accuracy if the $n$ samples were chosen randomly and not based on their scores. Figures in the lower row show the distribution of the critic score for train and test datasets. See Materials and methods for more details.
(TIF)

## Acknowledgments

Thanks to the Inria TAU team of the University of Paris-Saclay and the High Performance Computing Center of the University of Tartu for providing computational resources.

## Author Contributions

**Conceptualization:** Burak Yelmen, Aurélien Decelle, Guillaume Charpiat, Flora Jay.

**Data curation:** Burak Yelmen, Aurélien Decelle, Flora Jay.

**Formal analysis:** Burak Yelmen, Aurélien Decelle, Leila Lea Boulos, Antoine Szatkownik.

**Funding acquisition:** Flora Jay.

**Investigation:** Burak Yelmen, Aurélien Decelle, Leila Lea Boulos.

**Methodology:** Burak Yelmen, Aurélien Decelle, Cyril Furtlehner, Guillaume Charpiat, Flora Jay.

**Project administration:** Burak Yelmen, Guillaume Charpiat, Flora Jay.

**Resources:** Burak Yelmen, Aurélien Decelle, Flora Jay.

**Software:** Burak Yelmen, Aurélien Decelle, Leila Lea Boulos, Antoine Szatkownik, Flora Jay.

**Supervision:** Burak Yelmen, Guillaume Charpiat, Flora Jay.

**Validation:** Burak Yelmen, Aurélien Decelle, Antoine Szatkownik, Flora Jay.

**Visualization:** Burak Yelmen, Aurélien Decelle, Antoine Szatkownik.

**Writing – original draft:** Burak Yelmen.

**Writing – review & editing:** Burak Yelmen, Aurélien Decelle, Antoine Szatkownik, Cyril Furtlehner, Guillaume Charpiat, Flora Jay.

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
