## [Decision Letter · Decision Letter 0]

23 May 2023

Dear Yelmen,

Thank you very much for submitting your manuscript "Deep convolutional and conditional neural networks for large-scale genomic data generation" for consideration at PLOS Computational Biology.

As with all papers reviewed by the journal, your manuscript was reviewed by members of the editorial board and by several independent reviewers. In light of the reviews (below this email), we would like to invite the resubmission of a significantly-revised version that takes into account the reviewers' comments.

We cannot make any decision about publication until we have seen the revised manuscript and your response to the reviewers' comments. Your revised manuscript is also likely to be sent to reviewers for further evaluation.

Sincerely,

Piero Fariselli

Academic Editor

PLOS Computational Biology

William Noble

Section Editor

PLOS Computational Biology

Reviewer's Responses to Questions

**Comments to the Authors:**

Reviewer #1: This paper presents generative models for creation of synthetic genomes. Two variants of generative models are proposed (convolution WGAN and CRBM) and they are shown to perform better than the model previously developed in the group and a VAE. The synthetic genomes contain only SNPs and not other variations.

Major:

- The authors fail to convince the reader that artificial genomes (AGs) without larger variations (indels, rearrangements etc) are interesting and for what applications such AGs are useful. Such genomes are quite far from real ones. A title indicating that it is about synthetic SNP profiles would seem more appropriate.

- It is fairly straight-forward to simulate AGs with more conventional methods. The authors fail to explain why deep learning is needed. They should also compare to conventional algorithms. Here are some references:

X. Yuan, et al "An overview of population genetic data simulation", J. Comput. Biol., vol. 19, pp. 42-54, 2012.

A. Carvajal-Rodríguez, "Simulation of genomes: A review", Current Genomics, vol. 9, 2008.

Show in Context CrossRef Google Scholar

C. J. Hoggart et al., "Sequence-level population simulations over large genomic regions", Genetics, vol. 177, pp. 1725-1731, 2007.

Chun Li, Mingyao Li, "GWAsimulator: a rapid whole-genome simulation program", Bioinformatics, Volm24, pp 140–142, 2008

- The paper assumes deep familiarity with the earlier work in reference 20 to understand the results section. The data and models are not properly introduced and the reader needs to consult the methods section first to make any sense of it.

- It is argued that this work is important for reasons of GDPR, but there is no discussion of privacy preservation. How can we be certain that the original genomes are not identifiable if one has access to the model or a large number og AGs? It is hardly enough to say that "None of the models have produced identical sequences and no full sequences were copied from the training dataset" (line 59).

Reviewer #2: The paper presented an updated and extensive analysis of a previous detailed exploration regarding generative models for creating artificial genomes preserving real biological properties and structures. The article is clear, well-written, and thorough in its analysis. The paper certainly deserves a high consideration for publication. I would like to provide some minor considerations and comments that may help enhance its comprehensiveness and readability.

Here are a few points:

1. In their previous work, the authors calculated a measure of privacy loss introduced by Yale et al. https://inria.hal.science/hal-02160496/.

It would be interesting to include also in this paper a comparison table between the old and new models regarding privacy loss. It would be valuable to understand how the new models perform in terms of privacy score.

2. Generally, I would recommend reorganizing the results of the various models, along with their respective scores, like the AA score, in a table format. Alternatively, consider placing figure S10 in the results section. This would make it easier for readers to grasp the information immediately.

3. It would be beneficial to explicitly label the PCA densities in Figure 1a and 3a within the figures themselves, in addition to mentioning them in the figure legends. Similarly, for Figure 3c, including the label for LD within the figure itself would be helpful.

4. In the discussion section the authors better investigate the phenomenon of overfitting, specifically addressing the anomaly regarding the large difference between two terms of the AATS score (AAtruth and AAsyn). I propose relocating Table 1 to the results section, considering it presents new findings and it describe a new anomaly. Additionally, the analysis of the nearest neighbour chain should be presented in the materials and methods section.

5. In the methods section, the authors mention that during the training of the WCGAN, the train is stopped through a visual inspection of the overlap in PC space. While this method has its clear rationale, it seems quite arbitrary. I understand that it is beyond the scope of this paper to analyse potential alternatives in detail, but it would be interesting to discuss in the conclusion section any alternatives the authors consider feasible.

Reviewer #3: These authors present a novel approach based on Weierstrass GANs and convolutional processing for generating artificial genomes. They're building on their earlier work, while integrating several new ideas. Considerable effort is made to try to characterize the new networks, and to what extent they're realistic.

I am generally positive to the work, but I have noted some issues in presentation that I would like to see resolved:

1. The spans of the datasets are specified in terms of the number of SNPs. Some discussion is made of their distribution of minor allele frequencies, but very little with regards to their location (although their extraction from the Omni 2.5 panel is stated). It would make sense to state the total span in terms of physical or mapping distans along chromosome 1. From a population structure perspective, the actual length, rather than the number of variants, should be crucial in terms of the amount of underlying variation found in the genomes.

2. Color-coded scatter plots are used extensively in the main text and supplementary figures, with the exception of a few isobar plots. For example, Figure 1 a is using isobars, while 1 b is using scatter markers. With the very light transparency and high density in the center of these plots, it is very hard to even qualitatively judge whether the median, or mean, of the SNP distribution actually tracks the supposed center line or not. It should also be clarified whether the correlation metric stated is computed against the theoretical line, or a fitted version. After all, any linear correlation can end up with a high correlation coefficient, while we want to know if we reproduce y=x, not any y=kx+b. The presence of "label=" in some captions are distracting.

3. Along the same lines, in plots where multiple datasets are shown within the same plot, such as in Figure S11, it's problematic to not shuffle the order of the markers. By shuffling the z order in the center and rightmost panels in that plot, it would be more immediately clear to what extent the various distributions coincide. This applies to several other figures as well, including S7.

4. I think that there is reason to be cautious about the privacy guarantees, both from sets of generated artifical genomes, and especially if the underlying generating model would be distributed. Backpropagating attacks based on short haplotype excerpts, for example, could indicate whether a biological relative of an individual was memoized from the training set and reproduce larger parts of its genome, at least hypithetically. I find that the final paragraph in the discussion acknowledges this point to some extent, while the abstract, on the other hand, makes bolder claims regarding privacy.

5. While I understand the challenges involved, I find it unfortunate that no ablation study is performed. We rather see a discrete step between the old GAN approach and the new WGAN approach, while in fact there are multiple, distinct contributions that have been included.

6. What constitutes a good artificial genome can of course be hard to quantify. I find the LD plot S1 to be both convincing and concerning in this regard. Other possible metrics would be the reproduction of specific short k-mers (so not only maintaining pairwise LD, but identical specific short-range haplotypes). Since the WGAN critic has been trained, it could also be illuminating to at least verify that the critic fares worse at discriminating between WGAN AGs and real genomes, relative to VAE AGs vs. real genomes... or is the critic so inherently wed to the WGAN that it is blind to the drawbacks of the other methods?

**Have the authors made all data and (if applicable) computational code underlying the findings in their manuscript fully available?**

Reviewer #1: Yes

Reviewer #2: Yes

Reviewer #3: Yes

PLOS authors have the option to publish the peer review history of their article (what does this mean?). If published, this will include your full peer review and any attached files.

Reviewer #1: No

Reviewer #2: No

Reviewer #3: **Yes: **Carl Nettelblad
---

## [Decision Letter · Decision Letter 1]

28 Aug 2023

Dear Yelmen,

Thank you very much for submitting your manuscript "Deep convolutional and conditional neural networks for large-scale genomic data generation" for consideration at PLOS Computational Biology. As with all papers reviewed by the journal, your manuscript was reviewed by members of the editorial board and by several independent reviewers. The reviewers appreciated the attention to an important topic. Based on the reviews, we are likely to accept this manuscript for publication, providing that you modify the manuscript according to the review recommendations.

Sincerely,

Piero Fariselli

Academic Editor

PLOS Computational Biology

William Noble

Section Editor

PLOS Computational Biology

Reviewer's Responses to Questions

**Comments to the Authors:**

Reviewer #2: I thank the authors for their effort in their additional analyses and for satisfactorily responding to all my comments, especially adding the privacy loss score and performing membership inference attacks. The current version of the manuscript is definitely more complete and clearer in structure and exposition.

Reviewer #3: I appreciate the authors edits and their comments (including the extra critic score data in the reviewer responses). In general, I am satisfied with this state. I also find it crucial to show the slopes, since a slight deviation from the main diagonal would be almost imperceptible, but indicate a serious problem in e.g. the allele frequency plots.

My remaining thoughts are mainly also relating to figures and the presentation of data.

In Figure 3B, the y scale is different for the various plots in the lower set of subpanels in subfigure b. This is arguably due to outliers from some of the methods, but this artifact makes visual assessment of the data presented harder. When pairs of methods are compared, such as in Figure 5 b, I still think that it could make sense to join them in a single plot, IF shuffled z-ordering is implemented. I don't agree with the authors that this makes the result more visually confusing. If it is confusing, it's related to the nature of the overlap between the distributions. Such misleading covering of distributions is also clearly present in current supplementary Figure S3, where the RBM dominates the diagonal partially due to being drawn last. I think a Figure like S9 would also be more informative if the scatter point clouds were presented in a single figure.

This stackexchange thread gives a superficial illustration of the issue I'm referring to https://stats.stackexchange.com/questions/11984/how-can-i-remove-the-z-order-bias-of-a-coloured-scatter-plot (see especially the response from user bluenote10).

**Have the authors made all data and (if applicable) computational code underlying the findings in their manuscript fully available?**

Reviewer #2: Yes

Reviewer #3: Yes

PLOS authors have the option to publish the peer review history of their article (what does this mean?). If published, this will include your full peer review and any attached files.

Reviewer #2: No

Reviewer #3: No

Figure Files:

Data Requirements:

Reproducibility:

References:

---

## [Decision Letter · Decision Letter 2]

9 Oct 2023

Dear Yelmen,

We are pleased to inform you that your manuscript 'Deep convolutional and conditional neural networks for large-scale genomic data generation' has been provisionally accepted for publication in PLOS Computational Biology.

Best regards,

Piero Fariselli

Academic Editor

PLOS Computational Biology

William Noble

Section Editor

PLOS Computational Biology

Reviewer's Responses to Questions

**Comments to the Authors:**

Reviewer #3: I appreciate the changes made by the authors. While no work is ever complete, I think that the current version accuraretly reflects the work by the authors.

**Have the authors made all data and (if applicable) computational code underlying the findings in their manuscript fully available?**

Reviewer #3: Yes

PLOS authors have the option to publish the peer review history of their article (what does this mean?). If published, this will include your full peer review and any attached files.

Reviewer #3: **Yes: **Carl Nettelblad

---

## [Editor Report · Acceptance letter]

25 Oct 2023

PCOMPBIOL-D-23-00591R2 

Deep convolutional and conditional neural networks for large-scale genomic data generation

Dear Dr Yelmen,

I am pleased to inform you that your manuscript has been formally accepted for publication in PLOS Computational Biology. Your manuscript is now with our production department and you will be notified of the publication date in due course.

With kind regards,

Dorothy Lannert
